# The Effect of Maternal Coagulation Parameters on Fetal Acidemia in Placental Abruption

**DOI:** 10.3390/jcm11247504

**Published:** 2022-12-18

**Authors:** Atsuko Sugimoto, Tomohito Tanaka, Keisuke Ashihara, Atsushi Daimon, Misa Nunode, Yoko Nagayasu, Daisuke Fujita, Akiko Tanabe, Hideki Kamegai, Kohei Taniguchi, Kazumasa Komura, Masahide Ohmichi

**Affiliations:** 1Department of Obstetrics and Gynecology, Educational Foundation of Osaka Medical and Pharmaceutical University, 2-7 Daigakumachi, Takatsuki 569-8686, Japan; 2Translational Research Program, Educational Foundation of Osaka Medical and Pharmaceutical University, 2-7 Daigakumachi, Takatsuki 569-8686, Japan; 3Department of Obstetrics and Gynecology, Ikeda City Hospital, 3-Chome Jonan, Ikeda 563-0025, Japan; 4Department of Obstetrics and Gynecology, Suita Saiseikai Hospital, 1-2 Kawazonocho, Suita-City 564-0013, Japan

**Keywords:** fibrinogen, placental abruption, fetal acidemia

## Abstract

This study aimed to identify factors predicting the probability of serious fetal acidemia at delivery in placental abruption. We identified 5769 women who delivered at >22 weeks’ gestation at two institutions in a tertiary referral unit specializing in neonatal infant care between January 2007 and December 2011. Ninety-one abruption cases were identified based on clinical and histological diagnoses. Serious fetal acidemia was defined as a pH < 7.0 in the umbilical arterial blood at delivery. Using a linear discriminant function, we calculated the score to determine the probability of serious fetal acidemia. Serious fetal acidemia was observed in 34 patients (37.4%). A logistic regression model showed that abnormal fetal heart rate patterns (bradycardia and late decelerations), uterine spasm, and maternal plasma concentration of fibrinogen less than 288 ng/dL were significantly associated with the occurrence of serious fetal acidemia. We suggest that the implementation of maternal fibrinogen in patients with placental abruption is a prognostic factor for serious fetal acidemia at delivery.

## 1. Introduction

Placental abruption complicates approximately 0.5–1% of births [1,2]. Abruption is associated with serious and life-threatening obstetric complications in mothers, fetuses, and neonates [3]. The clinical presentation sometimes varies from asymptomatic, diagnosed only by placental examination at delivery, to severe abruption leading to fetal death and maternal morbidity [4]. The classic symptoms of abruption include vaginal bleeding, abdominal pain, uterine contraction, and tenderness [5]. The usefulness of ultrasonography [6] and a monitoring of fetal heart rate (FHR) has been reported as adjunctive diagnoses. Under the reassuring condition of the mother and fetus at preterm gestational ages with partial placental abruption, the patient may be managed conservatively to avoid morbidity due to immaturity [7,8]. On the other hand, the progression of abruption leads to the onset of maternal disseminated intravascular insemination (DIC) and fetal acidemia; therefore, the proper delivery timing is always difficult for many clinicians.

The important pathophysiologic factor is hemorrhage into the decidua basalis. The decidua is separated by this hemorrhage, leaving a thin layer adhered to the placenta. Therefore, placental abruption is often related to large amount of blood loss, either in the form of vaginal bleeding or concealed hemorrhage in the uterus. As with hemorrhagic shock, procoagulant substances (tissue thromboplastins) are released into the maternal circulation, and the coagulation cascade is extensively activated [9]. Simultaneously, a decrease in the placental vascular bed causes fetal hypoxia due to a reduction in blood flow. Fetal acidemia is defined as metabolic acidosis in the fetal umbilical cord’s arterial blood obtained at delivery (pH < 7.0, base deficit ≥ 12 mmol/L), which is an objective measure of intrapartum hypoxia-ischemia and has been correlated with the occurrence of hypoxic-ischemic encephalopathy (HIE) [10]. Prompt delivery is indicated before the irreversible progression of maternal DIC and fetal acidemia. It has been reported that the decreased fibrinogen level before delivery can accompany with adverse outcomes of infants in placental abruption. [11]. However, it remains unclear whether maternal coagulation parameters prior to delivery can be predictors of poor fetal prognosis with placental abruption. This study aimed to identify factors predicting the probability of serious fetal acidemia at delivery in placental abruption.

## 2. Materials and Methods

### 2.1. Study Subjects

Medical records of pregnant women admitted for delivery between 24 and 42 weeks of gestation at the Educational Foundation of Osaka Medical and Pharmaceutical University and Suita Saiseikai Hospital between January 2007 and December 2011 were carefully reviewed retrospectively as the test set. Those at the Educational Foundation of Osaka Medical and Pharmaceutical University, between January 2012 and December 2018, were reviewed as the validation set. Placental abruption was determined, as described in the next section.

Gestational age was determined on the basis of the mother’s last menstrual period and confirmed by ultrasound screening examination performed in the first trimester at 11–13 weeks of gestation. Women currently taking anticoagulants were excluded from the analysis. Informed consent was obtained from the patients as an opt-out on the website. Patients who refused to participate were excluded from the study. This study was approved by the Educational Foundation of the Osaka Medical and Pharmaceutical University Research Review Board (Assurance Number 2022-131).

### 2.2. Diagnosis of Placental Abruption

The diagnosis of placental abruption was made primarily on the basis of one or more of the following clinical findings: (1) vaginal bleeding; (2) spasm of uterus typified by the symptoms of abdominal pain and uterine tenderness; (3) abnormal ultrasonographic findings typified by increased heterogeneous placental thickness or retroplacental hematoma; (4) presence of abnormal FHR tracing [5,6]. The diagnosis was made by the presence of one or both of the following signs: (1) postpartum retroplacental hematoma at delivery; (2) intrauterine hematoma of Couvelaire uterus identified during cesarean section.

Women with the normal placenta and chronic abruption were excluded from this study. Abnormal FHR patterns were defined as persistent tracing of Category II or Category III FHR according to the clinical management guidelines developed by the American College of Obstetricians and Gynecologists (ACOG) Committee [12].

The maternal blood loss at delivery was examined, which means the amount of bleeding from intrapartum to the fourth stage of labor in cases of vaginal delivery. In cases of cesarean section, it means total estimated blood loss during surgery.

### 2.3. Biospecimen Collection and Processing

In cases of vaginal delivery, the maternal blood samples were collected during the first or second stage of labor. In cases of emergency cesarean section, the maternal blood samples were obtained from the decision of emergency cesarean section to the start of operation. The time span from the maternal sample collection to the fetal umbilical cord arterial blood obtain was less than 3 h in most cases. The maternal blood samples were immediately centrifuged. Coagulation and fibrinolytic system parameters, prothrombin tine (PT), fibrine/fibrinogen degradation products (FDP), and fibrinogen levels were analyzed using the STACIA system (LSI Medience, Tokyo, Japan). Platelet counts were measured using the fluorescence method on an NX9100 hematology analyzer (Sysmex, Kobe, Japan). During delivery, the umbilical cord was clamped twice, and arterial blood (0.2 mL) was collected on ice for gas analysis.

### 2.4. Diagnosis of Fetal Acidemia

Fetal acidemia was defined as metabolic acidosis in the fetal umbilical arterial blood collected at delivery (pH < 7.0 and base deficit ≥ 12 mmol/L). It is an objective measure of hypoxia-ischemia during labor and correlated with the development of cerebral palsy [10,13].

### 2.5. Statistical Analysis

Statistical calculations were performed using the JMP software package (version 15.1.1) (SAS Institute Japan, Tokyo, Japan) and R statistical software package 3.1.2 (R Foundation for Statistical Computing, Vienna, Austria; http://www.r-project.org (accessed on 5 August 2022). Continuous variables are expressed as the mean ± standard deviation. The Mann-Whitney U-test was used to compare continuous variables, and Fisher’s exact test was used to compare frequencies. The odds ratios with 95% confidence intervals (CI) were calculated for the identification of the risk factors for fetal acidemia. We conducted multivariate logistic regression analyses using all prognostic factors in the model and using a stepwise selection method in which terms were retained if they reached the significance level of 0.05. Receiver operating characteristic (ROC) curves were calculated to assess the relationship between the sensitivity and false-positive rate (1-specificity) for predictors of fetal acidemia. The optimal cutoff point was considered as the point corresponding to the highest sensitivity in relation to the highest specificity. Using logistic regression models, confounding effects was evaluated and discriminant function was calculated. A *p*-value of <0.05 was considered to be statistically significant.

We created a logistic model to estimate fetal acidemia. We then set the score to facilitate diagnosis from the model.

## 3. Results

Based on 5769 deliveries at the research time, abruption was observed in 1.80% (*n* = 104). Three cases of multiple pregnancies and 10 cases of intrauterine fetal death on admission were excluded, leaving 91 patients in the test set. Demographic and obstetric characteristics at delivery are summarized in Table 1. Among 91 patients with placental abruption, fetal acidemia was observed in 34 infants (37.4%). Most factors, including maternal age, parity, pre-pregnancy body mass index, hypertensive disorders of pregnancy (HDP), pre-eclampsia, cesarean section, gestational age, and birth weight, were not significantly different between the fetal acidemia and ‘no acidemia’ groups. Blood loss was significantly higher in the fetal acidemia group than in the ‘no acidemia’ group (*p* < 0.01).

Factors representing the severity of abruption affecting fetal acidemia were compared between the groups (Table 2). The odds ratio (95% CI) was 0.61 (0.21–1.77) in vaginal bleeding, 2.27 (0.97–5.41) in abnormal ultrasonographic findings, 20.48 (4.47–93.77) in abnormal FHR patterns, 8.62 (3.19–23.25) in uterine spasm, 1.22 (0.75–1.98) in gestational age at delivery (<34 weeks), 14.82 (5.22–42.03) in fibrinogen, 10.20 (3.59–29.02) in FDP, 3.33 (1.32–8.42) in platelet, and 4.73 (1.21–18.47) in PT, respectively.

The ROC analysis was performed to examine the predictive performance of maternal coagulation factors for fetal acidemia in patients with abruption (Figure 1). The optimal cutoff values were identified to be 288 ng/dL for fibrinogen, 31.0 µg/dL for FDP, 13.8 × 10^4^/µL for platelet, and 13.4 s for PT in the prediction of abruption in the patients. For these cutoff values, the area under the ROC curve (AUC) of the maternal fibrinogen was 0.850 (95% CI:0.743–0.918), which was significantly higher than that of FDP (0.787, 95% CI:0.666–0.873, *p* = 0.039), platelet (0.643, 95% CI:0.512–0.755, *p* = 0.002), and PT (0.748, 95% CI:0.539–0.883, *p* = 0.016).

Multiple logistic regression analysis was performed to predict fetal acidemia (Table 3). Abnormal FHR and uterine spasm were on the continuous scales, and maternal fibrinogen, FDP, Platelet, and PT were on the nominal scale (event No (0), Yes (1)). The odds ratio of primary outcomes was significantly higher in the abnormal FHR patterns (8.59, 95% CI 1.08–67.99; *p* = 0.0417), the uterine spasm (5.68, 95% CI 1.13–28.52; *p* = 0.0351), and fibrinogen (21.83, 95% CI 1.77–269.72, *p* = 0.0162), whereas there was no significant difference in the odds ratio in the gestational age at delivery, FDP, platelet, and PT.

Using the forward stepwise multiple logistic regression method, the reduced odds of fetal acidemia in the intervention group remained significant after adjustment for the other three risk factors (Table 4). We obtained the following logistic model:

Log{Pr/(1 − Pr)} = −3.597 + 1.442 × Uterine spasm (0, no; 1, yes) + 2.247 × abnormal FHR patterns (0, no; 1, yes) + 2.057 × maternal fibrinogen (0, ≥288 ng/dL; 1, <288 ng/dL).

However, the computation of the logistic model is complex and cannot be easily used in clinical settings. Therefore, we replaced the coefficient value of the estimated formula with a score. The simple formula was calculated and found to be

Diagnostic score = 1× uterine spasm (0, no; 1, yes) + 2 × abnormal FHR patterns (0, no; 1, yes) + 2 × maternal fibrinogen (0, ≥288 ng/dL; 1, < 288 ng/dL).

This diagnostic score was named the ‘predictive fetal acidemia in abruption score’ (PFAAS). The AUC was 0.893, and the optimal cutoff value for PFAAS was 3, as shown in Figure 2a. PFAAS ≥ 3 had a high sensitivity (94.1%, 95% CI 80.3–99.1), specificity (86.0%, 95% CI 74.2–93.7), positive predictive value (80.0% 95% CI 64.4–90.9), and negative predictive value (96.1, 95% CI 86.5–99.4) to predict fetal acidemia in abruption (Figure 2b).

Among the 2929 deliveries during the study period, there were 30 patients with placental abruption in the validation set. PFAAS was performed on these patients. Demographic and obstetric characteristics at delivery are summarized in Table 5. Table 6 shows the results of the PFAAS validation set. The sensitivity, specificity, positive predictive value, and negative predictive value were 100.0%, 68.2%, 53.3%, and 100.0%, respectively.

## 4. Discussion

In this study, we retrospectively investigated whether the maternal coagulation profile at delivery could predict fetal acidemia in placental abruption. The results showed that abnormal fetal heart rate patterns (bradycardia and late decelerations), uterine spasm, and maternal plasma fibrinogen concentration < 288 ng/dL were significantly related to the occurrence of fetal acidemia. We also found that PFAAS ≥ 3 were associated with fetal acidemia.

Matsuda et al. created a severe abruption score (SAS) to predict fetal acidemia in patients with placental abruption. They assessed 222 patients with placental abruption and found 43 cases of fetal acidemia. SAS was defined as A + 2B + 2C + 2D + 7E + 10F, where A was vaginal bleeding (0, no; 1, yes), B was gestational age less than 35 weeks (0, no; 1, yes), C was abdominal pain (0, no; 1, yes), D was abnormal ultrasonographic findings (0, no; 1, yes), E was persistent late decelerations (0, no; 1, yes), and F was bradycardia (0, no; 1, yes). Using SAS with a cutline of 11, the sensitivity, specificity, NPV, and PPV were 83.7%, 78.2%, 43.6%, and 93.8%, respectively. The most important factor was bradycardia, with an odds ratio of 50.34 [14]. In our study, the rates were 94.1%, 86.0%, 80.0%, and 96.1%, respectively. An abnormal FHR pattern was also an important factor, with an added ratio of 20.48. Kasai et al. evaluated the fetal outcomes in patients with placental abruption. They focused on vaginal and abdominal pain. They concluded that neonatal outcomes were significantly poorer in patients with concealed abruption, defined as abruptio placentae without vaginal bleeding and abdominal pain, than in those without them [15].

Several authors have focused on the coagulation profile of DIC associated with placental abruption. Fibrinogen, which is synthesized in the liver, is a protein with coagulation functions. D-dimer is a degradation product produced by the action of fibrinolytic enzymes on cross-linked fibrin. It is reasonable to assess the coagulation profile to predict the severity of placental abruption because patients with placental abruption have varying degrees of DIC. Jingjing et al. assessed 126 patients with placental abruption. Placental abruption severity was divided into grades I–III. They concluded that the pre-delivery volume of antepartum hemorrhage, D-dimer, FDP, and fibrinogen levels were correlated with placental abruption. There were differences in prenatal fibrinogen levels between patients with different placental abruption severities; however, there was also a significant nonlinear correlation [16]. Wang et al. evaluated the fibrinogen levels in 61 patients with placental abruption. They found that fibrinogen levels of <250 mg/dl were associated with fetal acidemia. The sensitivity, specificity, and PPV were 96.4%, 52.9%, and 77.1%, respectively. They also found that fibrinogen levels < 184 mg/dl were associated with low Apgar scores, and those less than 119 mg/dl were associated with stillbirth. They concluded that pre-delivery fibrinogen levels could predict adverse maternal and neonatal outcomes with placental abruption [11].

Our findings in the present study should be considered in light of several major limitations. First, the sample size was relatively small for multivariate analysis. Second, this study consisted of only East Asians; therefore, it is unclear whether our results apply to other ethics groups. Third, the timing of maternal blood sampling may limit predictability, especially if it is taken in the second stage of labor. Fourth, although the severity of the placenta abruption and the antepartum hemorrhage is related to the maternal coagulation parameters, also may related to the fetal acidemia, we did not investigate the antepartum hemorrhage in this study. Fifth, the amount of bleeding at delivery may not have been accurately counted, and the postpartum hemorrhage after the fourth stage of labor was not included in this study. Therefore, we recommend that the results of this study be confirmed by further investigation.

## 5. Conclusions

In conclusion, abnormal fetal heart rate patterns (bradycardia and late decelerations), uterine spasm, and maternal plasma fibrinogen concentration < 288 ng/dL were significantly related to the occurrence of fetal acidemia. Moreover, PFAAS ≥ 3 is associated with fetal acidemia.

## Figures and Tables

**Figure 1 jcm-11-07504-f001:**
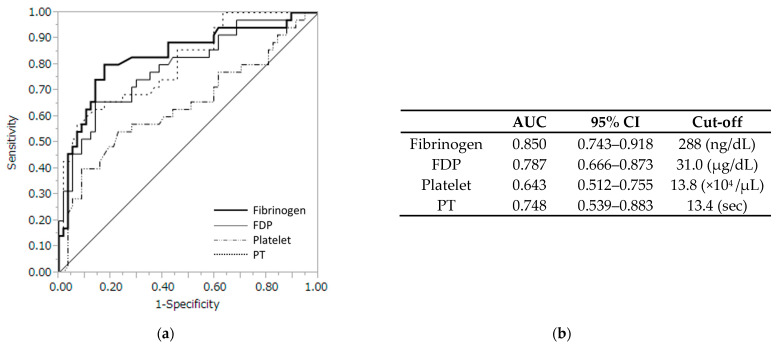
(**a**) Receiver operating characteristic curve with several coagulation profiles. (**b**) Fibrinogen levels with a cutoff of 288 ng/dL had the highest accuracy with an AUC of 0.85 (95% CI, 0.743–0.918). AUC, area under the curve; CI, confidence interval; FDP, fibrin/fibrinogen degradation products; PT, prothrombin time.

**Figure 2 jcm-11-07504-f002:**
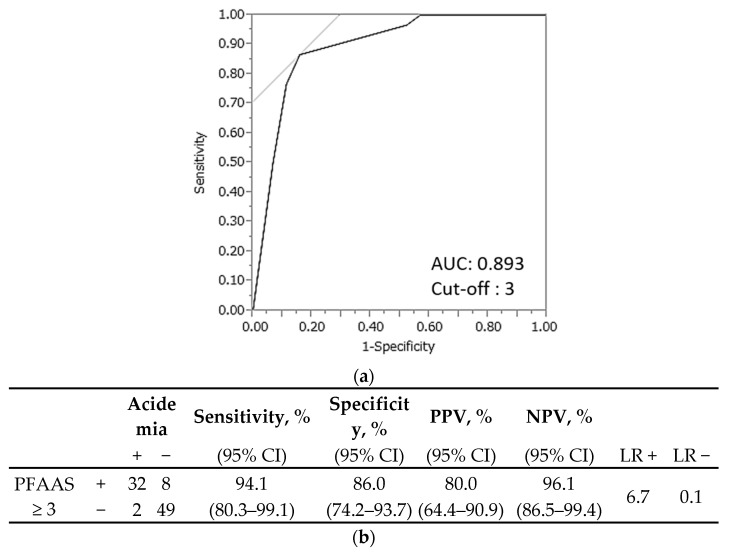
Prediction of fetal acidemia in placental abruption using the predictive fetal acidemia in abruption score (PFAAS) in the test set. (**a**) The area under the curve was 0.893, and the optimal cutoff value for the PFAAS was 3. (**b**) The sensitivity, specificity, positive predictive value, and negative predictive value were 94.1%, 86.0%, 80.0%, and 96.1%, respectively. CI, confidence interval; PPV, positive predictive value; NPV, negative predictive value; LR, likelihood ratio.

**Table 1 jcm-11-07504-t001:** Demographics and obstetrical characteristics of the study participants.

	Acidemia (*n* = 34)	No Acidemia (*n* = 57)	*p*-Value
Maternal age * (years)	31.9 ± 5.8	32.1 ± 5.1	0.803
<25	2 (5.9%)	4 (7.0%)	0.993
26–34	22 (64.7%)	37 (64.9%)	
35–39	8 (23.5%)	15 (26.3%)	
≥40	2 (5.9%)	3 (5.3%)	
Parity (*n*)			0.399
Primiparous	15 (41.2%)	32 (56.1%)	
Parity = 1	10 (29.4%)	16 (28.1%)	
Parity ≥ 2	9 (26.5%)	9 (15.8%)	
Pre-pregnancy BMI *	20.8 ± 1.9	20.5 ± 2.5	0.526
HDP (*n*)	6 (17.6%)	6 (10.5%)	0.331
Pre-eclampsia (*n*)	5 (13.5%)	6 (10.5%)	0.554
Cesarean section (*n*)	32 (94.1%)	49 (86.0%)	0.229
GA at delivery * (weeks)	33.8 ± 4.3	34.4 ± 3.3	0.882
22–27	6 (17.6%)	5 (8.8%)	0.191
28–33	8 (23.5%)	14 (24.6%)	
≥34	20 (58.8%)	38 (66.7%)	
Birth weight * (g)	1983 ± 749	2176 ± 711	0.111
Blood loss * (mL)	1840 ± 1148	1032 ± 580	0.0001

* Mean ± standard deviation (SD). BMI, body mass index; HDP, hypertensive disorders of pregnancy; GA, gestational age.

**Table 2 jcm-11-07504-t002:** Results of univariate analysis in terms of prediction factors of fetal acidemia.

	Acidemia (*n* = 34)	No Acidemia (*n* = 57)	OR (95% CI)
Vaginal bleeding	26 (76.5%)	48 (84.2%)	0.61 (0.21–1.77)
Abnormal ultrasonographic findings	20 (58.8%)	22 (38.6%)	2.27 (0.97–5.41)
Abnormal FHR patterns	32 (94.1%)	25 (43.9%)	20.48 (4.47–93.77)
Uterine spasm	21 (61.8%)	9 (15.8%)	8.62 (3.19–23.25)
Gestational age at delivery (<34 weeks)	12 (35.3%)	15 (26.3%)	1.22 (0.75–1.98)
Maternal coagulation parameters			
Fibrinogen (ng/dL)	170.5 ± 120.0	334.2 ± 96.6	14.82 (5.22–42.03)
FDP (µg/dL)	186.2 ± 272.5	19.4 ± 37.9	10.20 (3.59–29.02)
Platelet (×10^4^/µL)	16.0 ± 6.7	19.8 ± 6.0	3.33 (1.32–8.42)
PT (second)	16.2 ± 6.2	12.3 ± 1.8	4.73 (1.21–18.47)

OR, odds ratio; CI, confidence interval; FHR, fetal heart rate; FDP, fibrin/fibrinogen degradation products; PT, prothrombin time.

**Table 3 jcm-11-07504-t003:** Results of multiple logistic regression analysis in terms of prediction of fetal acidemia.

	OR (95% CI)	*p*-Value
Abnormal FHR patterns	8.59 (1.08–67.99)	0.0417
Uterine spasm	5.68 (1.13–28.52)	0.0351
Gestational age at delivery (<34 weeks)	0.54 (0.06–6.53)	0.6301
Maternal coagulation parameters		
Fibrinogen (<288 ng/dL)	21.83 (1.77–269.72)	0.0162
FDP (>31.0 µg/dL)	1.15 (0.13–9.93)	0.9000
Platelet (<13.4 × 10^4^/µL)	3.34 (0.4–28.01)	0.2655
PT (>13.4 s)	0.72 (0.51–1.02)	0.0680

OR, odds ratio; CI, confidence interval; FHR, fetal heart rate; FDP, fibrin/fibrinogen degradation products; PT, prothrombin time.

**Table 4 jcm-11-07504-t004:** Results of multivariate analysis in terms of risk factors of fetal acidemia.

	OR (95% CI)	*p*-Value
Abnormal FHR patterns	9.46 (1.71–52.28)	0.0100
Uterine spasm	4.23 (1.07–16.73)	0.0399
Maternal coagulation parameters		
Fibrinogen (<288 ng/dL)	7.82 (2.09–29.30)	0.0023

OR, odds ratio; CI, confidence interval; FHR, fetal heart rate.

**Table 5 jcm-11-07504-t005:** Demographics and obstetrical characteristics of the validation set.

	Acidemia (*n* = 8)	No Acidemia (*n* = 22)	*p*-Value
Maternal age * (years)	33.9 ± 3.9	32.0 ± 4.8	0.331
<25	0 (0%)	1 (4.5%)	0.119
26–34	3 (37.5%)	15 (68.2%)	
35–39	5 (62.5%)	4 (18.2%)	
≥40	0 (0%)	2 (9.1%)	
Parity (*n*)			0.932
Primiparous	4 (50.0%)	10 (45.4%)	
Parity = 1	3 (37.5%)	8 (36.4%)	
Parity ≥ 2	1 (12.5%)	4 (18.2%)	
Pre-pregnancy BMI *	21.6 ± 2.6	21.2 ± 2.4	0.688
HDP (*n*)	2 (25.0%)	2 (9.1%)	0.257
Pre-eclampsia (*n*)	2 (25.0%)	0 (0%)	0.015
Cesarean section (*n*)	8 (100%)	22 (100%)	
GA at delivery * (weeks)	32.1 ± 4.1	33.3 ± 4.2	0.507
22–27	1 (12.5%)	2 (9.1%)	0.711
28–33	4 (50.0%)	8 (36.4%)	
≥34	3 (37.5%)	12 (54.5%)	
Birth weight * (g)	1612 ± 608	1972 ± 765	0.242
Blood loss * (mL)	1008 ± 314	887 ± 383	0.434

* Mean ± standard deviation (SD). BMI, body mass index; HDP, hypertensive disorders of pregnancy; GA, gestational age.

**Table 6 jcm-11-07504-t006:** Prediction of fetal acidemia in placental abruption using the predictive fetal acidemia in abruption score (PFAAS) in the validation set. The sensitivity, specificity, positive predictive value, and negative predictive value were 100.0%, 68.2%, 53.3%, and 100.0%, respectively. CI, confidence interval; PPV, positive predictive value; NPV, negative predictive value; LR, likelihood ratio.

		Acidemia		Sensitivity, %	Specificity, %	PPV, %	NPV, %		
		+	−	(95% CI)	(95% CI)	(95% CI)	(95% CI)	LR +	LR −
PFAAS ≥ 3	+	8	7	100.0	68.2	53.3	100.0	3.1	0.0
	−	0	15	(51.8–100.0)	(45.1–86.1)	(26.6–78.7)	(69.8–100.0)		

## Data Availability

The data presented in this study are available upon request from the corresponding author.

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
