# Peer review of "The Effect of Maternal Coagulation Parameters on Fetal Acidemia in Placental Abruption"

_jcm, 2022, doi:10.3390/jcm11247504_

Round 1

Reviewer 1 Report

It’s a very interesting single-center study about maternal coagulation parameters in predicting fetal acidemia during placenta abruption. I have some comments and remarks regarding the manuscript.

(1)   In the methods, author said the maternal blood samples were obtained at delivery, here you mean the sample was collect during the first or second stage of labor? If the blood sample was from first stage, what’s the time span from the maternal sample collection to the fetal umbilical cord arterial blood obtain?

If the maternal blood sample was from the second stage of labor, predictive effect will be limited.

(2)   As we know, the severity of the placenta abruption and the antepartum hemorrhage is related to the maternal coagulation parameters, also may related to the fetal acidemia, these should be considered as obstetrical characteristics in the study.

(3)   In table 1, the blood loss has significant difference between acidemia and no acidemia, but what the “blood loss” mean was not clarified in the methods. Blood loss means antepartum or postpartum hemorrhage?

(4)   The difference of FHR tracing (Category II or Category III) should also be considered as obstetrical characteristic.

Author Response

We appreciate the time and effort provided by the editor and referees in reviewing our manuscript. We have addressed all issues indicated in the review report and hope that the revised version meets the journal's requirements for publication.

Response to Comments from Reviewer 1:

Comment 1:   In the methods, author said the maternal blood samples were obtained at delivery, here you mean the sample was collect during the first or second stage of labor? If the blood sample was from first stage, what’s the time span from the maternal sample collection to the fetal umbilical cord arterial blood obtain?

If the maternal blood sample was from the second stage of labor, predictive effect will be limited.

Response: In cases of vaginal delivery, the maternal blood samples were collected during the first or second stage of labor. In cases of emergency cesarean section, the maternal blood samples were obtained from decision of emergency cesarean section to the start of operation. The time span from the maternal sample collection to the fetal umbilical cord arterial blood obtain was less than 3 hours in most cases. According to your suggestions, we added the sentences about this in method section. (page 3, line 103-109)

We also added the limitation. (page 8, line 268-269)

Comment 2:     As we know, the severity of the placenta abruption and the antepartum hemorrhage is related to the maternal coagulation parameters, also may related to the fetal acidemia, these should be considered as obstetrical characteristics in the study.

Response: Unfortunately, we could not make sure of them. Our institution is Advanced treatment hospital; most patients with placental abruption were sent from other hospital. Thus, it was impossible to predict the amount of antepartum hemorrhage accurately. According to your suggestion, we added the sentences about this in limitation as below. (page 8, line 269-272)

“Although the severity of the placenta abruption and the antepartum hemorrhage is related to the maternal coagulation parameters, also may related to the fetal academia, we did not investigate the antepartum hemorrhage in this study.”

Comment 3:    In table 1, the blood loss has significant difference between acidemia and no acidemia, but what the “blood loss” mean was not clarified in the methods. Blood loss means antepartum or postpartum hemorrhage?

Response: In cases of vaginal delivery, the blood loss means the amount of bleeding from intrapartum to the fourth stage of labor. In cases of cesarean section, it means total estimated blood loss during surgery. We added the sentences about “blood loss” in method section as below. (page 3, line 99-101)

“The maternal blood loss at delivery was examined, which means the amount of bleeding from intrapartum to the fourth stage of labor in cases of vaginal delivery. In cases of cesarean section, it means total estimated blood loss during surgery.”

The following sentences were also added in the limitation. (page 8, line 272-274)

“The amount of bleeding at delivery may not have been accurately counted, and the postpartum hemorrhage after the fourth stage of labor was not included in this study.”

Comment 4:    The difference of FHR tracing (Category II or Category III) should also be considered as obstetrical characteristic.

Response: We defined the status of persistent Category II or Category III FHR tracing as abnormal FHR pattern. Unfortunately, data collection and evaluation were not performed separately for Category II and Category III.

Reviewer 2 Report

Dear Authors, 

I would like to congratulate you on a fantastic paper. The paper has an appropriate study design and soundness of results. The score you have developed is much easier to remember and more applicable than other ones already published.

I don't have any negative comments on the paper. It is obvious that the study was done on the asian population and that the numbers taken into account are small, but that is unavoidable in small center studies. Nonetheless, it's a good study. 

Author Response

We appreciate the time and effort provided by the editor and referees in reviewing our manuscript. We have addressed all issues indicated in the review report and hope that the revised version meets the journal's requirements for publication.

Response to Comments from Reviewer 2:

Comment 1: I would like to congratulate you on a fantastic paper. The paper has an appropriate study design and soundness of results. The score you have developed is much easier to remember and more applicable than other ones already published.

I don't have any negative comments on the paper. It is obvious that the study was done on the asian population and that the numbers taken into account are small, but that is unavoidable in small center studies. Nonetheless, it's a good study. 

Response: Thank you for your kind advice on our manuscript. The sample size was relatively small in our study as you mentioned. We stated the limitation about this. I look forward to working with you to move this manuscript closer to publication in Journal of Clinical Medicine.